# UViM: A Unified Modeling Approach for Vision with Learned Guiding Codes

**Alexander Kolesnikov**[*][†]   **André Susano Pinto**[*][†]
**Lucas Beyer**[*]   **Xiaohua Zhai**[*]   **Jeremiah Harmsen**[*]   **Neil Houlsby**[*]

Google Research, Brain Team Zürich

{akolesnikov,andresp,lbeyer,xzhai,jeremiah,neilhoulsby}@google.com

## Abstract

We introduce UViM, a unified approach capable of modeling a wide range of computer vision tasks. In contrast to previous models, UViM has the same functional form for all tasks; it requires no task-specific modifications which require extensive human expertise. The approach involves two components: (I) a *base model* (feed-forward) which is trained to directly predict raw vision outputs, guided by a learned discrete code and (II) a *language model* (autoregressive) that is trained to generate the guiding code. These components complement each other: the language model is well-suited to modeling structured interdependent data, while the base model is efficient at dealing with high-dimensional outputs. We demonstrate the effectiveness of UViM on three diverse and challenging vision tasks: panoptic segmentation, depth prediction and image colorization, where we achieve competitive and near state-of-the-art results. Our experimental results suggest that UViM is a promising candidate for a unified modeling approach in computer vision.

## 1 Introduction

Many computer vision tasks require producing high-dimensional structured outputs. Examples include various types of image segmentation, monocular depth estimation, surface normal estimation, colorization, object detection, image super-resolution, etc. By handcrafting architectures and training procedures specific to each task, the structure of those target outputs can be exploited to learn better models. However, this fragmented approach impedes the ability to build a general solution ready to be applied to any task.

For the tasks above, that require predicting high-dimensional structured outputs, direct use of powerful parametric models such as CNNs [27, 44, 16] and Vision Transformers [9], trained with decomposable (e.g. pixel-wise) loss is not sufficient, as this basic approach lacks the ability to model the structure of the output. To address this shortcoming, standard approaches turn to using additional modeling components such as, for example, anchor boxes [37, 30], non-maximal suppression [37, 30], matching losses [2, 5, 6] or conditional random fields [3, 55].

Recently, there have been significant advances in the modeling of complex structured outputs in the context of language generation and (conditional) image generation: autoregressive models [49, 41, 25], GANs [13], VAE [22], VQVAE [51], diffusion models [45, 18]. However, using such techniques to tackle discriminative problems in a unified way remains under-explored.

In this work, we propose a new approach, UViM, capable of modeling many vision tasks, leveraging recent advances in discrete representation learning [51] and language modeling [52]. We show competitive results in three diverse tasks: panoptic segmentation [23], depth prediction [43] and colorization [57]. Crucially, there are no task-specific components required for each task. All of the tasks use the same model and are amenable to transfer learning from standard pre-trained models.

---

[*]Significant technical contribution. [†] Shared first authorship.

36th Conference on Neural Information Processing Systems (NeurIPS 2022).

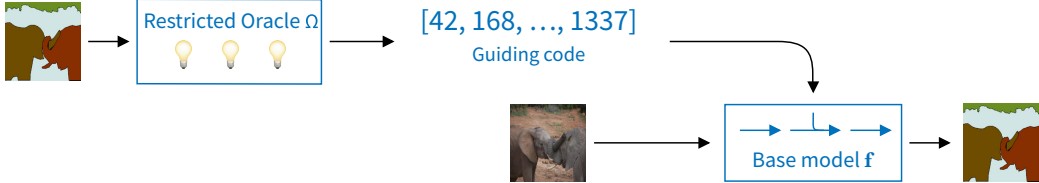

(a) **Stage I** training: we train the base model $f$, which is guided by the code produced by the *restricted oracle* model $\Omega$. The oracle has access to the ground-truth label, but is only allowed to communicate with $f$ by passing a short discrete sequence, which we call a *guiding code*.

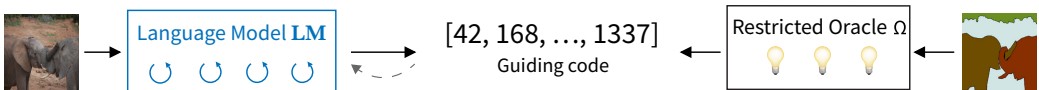

(b) **Stage II** training: we train a *language model* (LM) to output a *guiding code* by learning to mimic the oracle, but using only the image input.

Figure 1: An overview of the UViM learning procedure. Blue blocks depict parts of the model which are being optimized, while black blocks depict frozen components.

## 2 Unified Modeling Approach for Vision

We first discuss the motivation and inspiration behind our unified modeling approach: UViM. Then we give a high-level overview, followed by an in-depth explanation of its design and technical details.

### 2.1 Motivation

The field of computer vision made a huge leap by transitioning to models based on rich parametric functions (CNNs [27, 44, 48, 16], ViTs [9]). Combined with well-working gradient-based algorithms to train these functions (e.g. Adam [21]), it enables learning of complex feedforward mappings from inputs to outputs $f : X \rightarrow Y$, which we call *base models*.

Despite the ability to train powerful and reusable base models, different vision applications, particularly those involving high-dimensional structured outputs, such as object bounding boxes, per-pixel segmentation masks or 3D point clouds, still rely on highly customized components and techniques listed in the introduction.

The necessity for introducing these non-trivial custom components has the same underlying root cause: the outputs are high-dimensional and structured (interdependent) and modeling complex interactions is a necessary condition for succeeding at a given task. Modeling such data is a longstanding challenge in computer vision (and beyond), with numerous books on the subject [34, 26] and remains a relevant area of research.

In contrast to computer vision, a prominent unified modeling approach has been adopted in NLP. Many NLP tasks can be handled by an autoregressive sequence model [35] parameterized by the Transformer architecture [52]. This approach combines multiple desirable properties: it is theoretically sound, expressive (capable of modeling a joint probability distribution of the outputs) and there are robust techniques for training such models.

Why has the computer vision field not yet adopted a similar unified model? There are papers that demonstrate that the NLP-like approach, based on autoregressive sequence models, is viable for some image tasks [4, 38]. However, it only works for tasks that have compact output representations; the additional challenge in vision is that outputs are often very high-dimensional. NLP models typically model sequences of length up to 10'000 tokens, while in vision, outputs, such as per-pixel image segmentation/instance masks, may contain millions of elements. It is computationally prohibitive to apply autoregressive sequence models directly to such tasks.

## 2.2 Unified vision model via learned guiding code

Now we present our unified modeling approach for computer vision. We first give a high-level overview of the proposed model and then describe its components in detail.

We devise a unified vision model as a composition of a standard feedforward *base model* and an autoregressive *language model* of a short sequence. Our decomposition works well even for vision tasks that deal with extremely high dimensional and structured outputs.

Our key insight is to reduce the original task of modeling very high-dimensional structured output (e.g. panoptic segmentation mask) to modeling a short discrete sequence with the language model. For this, we propose an optimization procedure, illustrated in Figure 1. The resulting model during inference is depicted in Figure 2.

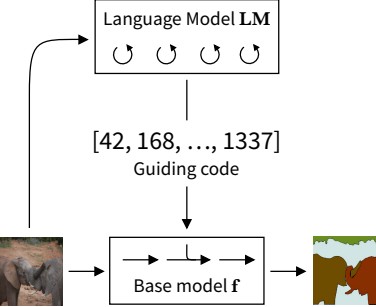

Figure 2: The schematic illustration of UViM during inference.

**Stage I training: learning with a guiding code.** To build a unified vision model we start from a base model $f : \mathcal{X} \to \mathcal{Y}$, which directly maps task inputs to its outputs. As discussed above, learning such model with simple element-wise loss for a structured label space $\mathcal{Y}$ does not result in a good prediction model, as it is not modeling complex interactions within the output space.

To compensate for this modeling deficiency, we introduce an input $z \in \mathcal{Z}$, called *guiding code*. The assumption is that given $x$ and $z$, the elements of the output $y$ have fewer dependencies, and can be modelled well by the *base model*. As an illustrative example, consider colorization: given a grayscale image of a car, the pixel colors are highly dependent (most cars are of uniform color). However, given a guiding code with the information "the car is red", such cross-pixel dependencies cease to exist.

The guiding code $z$ has two key properties. First, it is represented as a short discrete sequence of the fixed length $n$, i.e. $z = (z_1, z_2, \ldots, z_n)$. Second, it is derived from the output $y$ through the special function $\Omega$: $z = \Omega(y)$. We call $\Omega$ the *restricted oracle*, because it has access to the target (ground truth) $y \in \mathcal{Y}$, but at the same time is forced to compactly represent the information which will help $f$ to solve the task. Note, the restricted oracle is only used during training, but not at test time.

We train $f$ and $\Omega$ jointly and end-to-end by minimizing a reconstruction loss between $f(x, \Omega(y))$ and $y$. For the reconstruction loss, we use the simplest task-appropriate loss function, e.g. pixel-wise cross-entropy or mean squared error. See stage I training step illustrated in Figure 1(a).

Empirically, we observe that the function $f(x, z)$, "aided" by the guiding code from the restricted oracle, is capable to solve the complex vision tasks very well, as measured by the task-specific standard metrics. Note, that $f(x, z)$ is not a prediction model, as $z$ depends on the ground truth $y$. Nevertheless, in this stage we have introduced a crucial component, which helps to reduce a high-dimensional structured prediction task to modeling a short sequence of discrete variables $z$.

**Stage II training: learning to model the guiding code.** At the second stage, we model the discrete sequence $z$ using the input $x$. The training data is a collection of input-output pairs $(x, \Omega(y))$, where $\Omega$ is the fixed restricted oracle trained from the stage I. Note, that this task is equivalent to many standard NLP problems (except the input is an image) and there is a vast number of research and tools to tackle it. We use a standard encoder-decoder language model [52] $\text{LM} : \mathcal{X} \to \mathcal{Z}$, which processes the image through the encoder and passes it to the autoregressive decoder. Training is performed end-to-end with gradient-based optimization. See Figure 1(b) for illustration of stage II learning step.

**Resulting unified vision model.** As a result of the two-stage optimization procedure, we obtain a final model $f(x, \text{LM}(x))$, which we call UViM, short for *Unified Vision Model*. See Figure 2 for an overview. Later in the experimental section we show that such a model can be successfully trained to model highly structured outputs for very different vision tasks.

## 2.3 Implementation details

**Joint training of base model $f$ and restricted oracle $\Omega$.** Stage I training involves training a model that contains a discrete bottleneck $z = \Omega(y)$, which is used to guide the base model $f(x, z) \to y$. Such discrete bottleneck is problematic for training with gradient-based methods, as it does not have

a gradient. To address this, we employ the technique introduced by the seminal VQ-VAE paper [51]. The key idea is to map the embeddings to be quantized to the nearest entry in a dictionary of $N$ $d$-dimensional embeddings. We refer the reader to the paper for a detailed overview.

**Addressing embedding dictionary usage.** We observed that during Stage I training the usage of VQ-VAE dictionary may be highly unbalanced and certain entries going unused. To address this, we adapt the classic Linde-Buzo-Gray [32] splitting algorithm to VQ-VAE's dictionary learning procedure. Specifically, if, throughout the training process, we detect an unused embedding, we then take the most frequently used embedding and split it into two new embeddings by applying a tiny noise, and consequently replacing the unused one.

**Architectures of functions $f$, $\Omega$ and LM.** Throughout our experiments, we strive to use as uniform setup as possible. By default, we use a plain ViT architecture to parameterize all functions. Specifically, function $f$ and $\Omega$ are modeled by the ViT architecture introduced in [9]. For historical reasons we equip $\Omega$ with an additional input $x$, though later we verified that it does not affect the performance and can be omitted. The function LM is a standard encoder-decoder model and consists of two parts: $\text{LM}_{enc}$ and $\text{LM}_{dec}$. The encoder, $\text{LM}_{enc}$ is also modeled by the ViT backbone. The decoder, $\text{LM}_{dec}$ is modeled by the standard transformer decoder, which is identical to the ViT model without initial projection for image patches.

**Controlling sequence length of guiding code.** As $\Omega$ is parameterized by the ViT model, its output is a collection of vectors arranged as a grid. To disentangle the grid size from the guiding code size, we optionally perform a linear spatial resize operation. Appendix B explores the code's locality.

**Dropout for guiding code.** Empirically, we find that modeling the code $z$ during phase II can be quite challenging. This motivates us to explore a code dropout mechanism to affect the code complexity. For each training example in a batch, we randomly select an integer $k$ from 0 to $n$, where $n$ is the code length. Then, we set a random subset of $k$ codewords to 0 before inputting it to the model $f$. As a result, *base model* learns to not rely on any individual code too heavily and the code becomes more robust. Intuitively, we expect that this can help to get better final stage II UViM model. We empirically validate the effect of this approach in Section 4.

**Sampling from LM at test time.** Running UViM at test time involves evaluating two functions: $\text{LM} : \mathcal{X} \to \mathcal{Z}$ and then $f : \mathcal{X} \times \mathcal{Z} \to \mathcal{Y}$. While evaluating $f$ is straightforward, the function LM is autoregressive and models a joint distribution $p(z|x) = p(z_1, z_2, \ldots, z_n|x)$. Sampling from $p(z|x)$ is a known and extensively studied task in NLP literature [7, 47, 52]. In our initial experiments we observed that the the simplest sampling approach seems to work well and more complex sampling techniques, such as beam search are not necessary. Thus, we sample $z$ using the most standard coordinate-wise sequential sampling $z_k \sim p(z_k|z_{k-1} \ldots z_1, x)$. Note, we can optionally vary the temperature $T$ of the conditional distributions. By setting $T = 0$ we can produce the "most likely" sample $z$, but lose diversity. Contrary, with the default temperature $T = 1$, we can get diverse samples (and consequently diverse predictions), but potentially at the expense of prediction quality.

## 3   Experiments

We apply UViM to three diverse tasks: a general scene understanding panoptic segmentation task, a conditional generative image colorization task and a 3D scene understanding task of depth prediction. With UViM, we use a unified setup for all three seemingly different tasks. Quantitative results are presented in Table 1 and qualitative results are in Appendix A. We describe our main modeling choices below, however we also provide full configuration files (as-is) in the Appendix D. The full UViM code is publicly available in the `big_vision` codebase.[2]

**Experimental setup for stage I.** We parameterize the *base model* $f$ and the *restricted oracle* $\Omega$ with ViT-B/16 model. For $\Omega$ we use 6 layers instead of 12, as in the initial experiments we observed that a relatively small capacity is sufficient. Both models are trained from scratch.

The input and output resolution during stage I for all tasks is $512 \times 512$. For optimization we use a variant of Adafactor [42] introduced in [56]. Due to differences in dataset size, we tune the learning rate and number of epochs per task, but all other hyperparameters are the same.

For the guiding code, $z \in \mathcal{Z}$, produced by the restricted oracle, we use a sequence length of 256 with 4096 dictionary entries. To put this choice into perspective, for the panoptic task, the original panoptic

---

[2]`https://github.com/google-research/big_vision`

Table 1: Comparison of presented modeling approach (UViM) and other related works discussed in Section 5 including current state of the art. Note that ours is the only work covering a set of significantly different tasks dominated by different types of approaches. Standard deviations are computed across three independent reruns.

| COCO Panoptic [PQ] | | NYU Depth v2 [RMSE] | | ImageNet Colorization [FID-5k] | |
|---|---|---|---|---|---|
| UViM (ours) | $45.8 \pm 0.09$ | UViM (ours) | $0.467 \pm 0.009$ | UViM (ours) | $16.99 \pm 0.001$ |
| DETR-R101 [2] | 45.1 | DenseDepth [1] | 0.465 | COLTRAN [28] | 19.37 |
| Mask2Former [5] | **57.8** | BinsFormer [29] | **0.330** | Palette [40] | **15.78** |

mask is encoded as roughly $512 \cdot 512 \cdot 2 \approx 524\,000$ discrete values, each ranging approximately from 0 to 100. Thus, $z$ is more than three orders of magnitude more compact than the original label.

**Experimental setup for stage II.** The language model consists of the encoder and autoregressive decoder. For the encoder, by default, we use the ViT-L/16 model. We initialize the encoder with the ImageNet-21k [39] pre-trained model from [46]. For the decoder, we use the ViT-B model. Note, that there is no initial patch projection, as it uses guiding code $z$ as autoregressive input, this is equivalent to the standard BERT-Base [8] architecture.

As in the stage I, the input and output resolution for all tasks is $512 \times 512$, except for the panoptic task, where we use a higher input resolution of $1280 \times 1280$. For optimization, we use the same optimizer as in Stage I. For all tasks, we use a base learning rate of $0.001$ with cosine decay and, additionally, apply a 10-fold reduction for the pre-trained encoder weights. Due to differences in dataset size, the number of epochs is tuned per task.

For all our experiments we use Google Cloud TPU-v3 hardware. A phase I training run for panoptic segmentation requires 1.9k TPU-v3 hours, while a phase II training run requires 0.9k TPU-v3 hours.

**Data augmentations** We strive to use the simple and standard augmentations for all tasks. At train time we opt for using an inception crop [48], random horizontal flipping, followed by resize to a square-shaped input. At test time we only squared-shaped resize the inputs to the input resolution, with the exception of colorization which performs a center-crop before resizing for consistency with existing evaluation methodologies.

### 3.1 Panoptic segmentation

Panoptic segmentation [23] is a general scene understanding task, which requires mapping every image pixel to its semantic class and, if applicable, instance ID. We adopt the raw target representation used in the original paper: a 2-channel mask, where the first channel encodes semantics, and the second channel encodes instance IDs. During training we assign instances IDs in an a raster scan order of object centers.

We train on the COCO panoptic 2017 [31, 23] dataset. It has approximately 118'000 training images and 5'000 official validation images which we use for test. All hyper-parameters were selected on 4'096 images held out from the training data. For evaluation, we use the official metric, called panoptic quality (PQ), which jointly estimates the accuracy of semantic and instance segmentation. We train stage I for 1000 epochs and stage II for 200 epochs.

As the reconstruction loss during stage I, we use the standard cross-entropy categorical loss for each channel independently. At test time, the output mask is first formed by the predicted instance channel. Then each instance is labeled my the majority vote of pixels from the semantic channels. This avoids inconsistencies in which pixels with the same instance id, but different semantic categories are interpreted as different instances. We additionally remove tiny objects that occupy less than 0.1% of all pixels. At test time, we resize the outputs to the target resolution via nearest neighbour.

Table 1 shows that UViM achieves 45.8 PQ, outperforming a recent strong baseline model DETR-R101 [2]. We focus on evaluating the generality of our approach, hence we avoid specialization towards individual tasks, such as commonly-used feature pyramids or scale jitter augmentations. As a result we lag behind the most recent state-of-the-art [5]. We expect that the gap can be bridged by further refining UViM with better understanding of its components and smarter modeling choices.

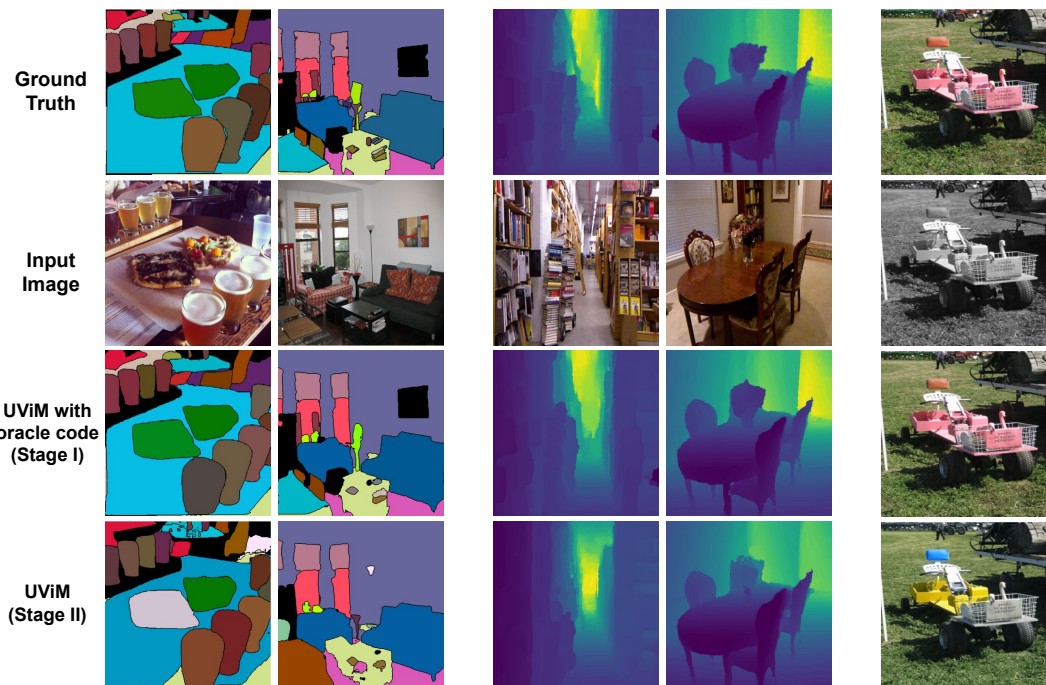

Figure 3: We demonstrate how UViM performs across three different diverse tasks. Note that when provided with oracle's *guiding code* it achieves near perfect results (3rd row). Predictions of the final UViM model are exemplified in 4th row. They are generally of very high quality and confirm that LM can successfully learn to produce the guiding code from the image input.

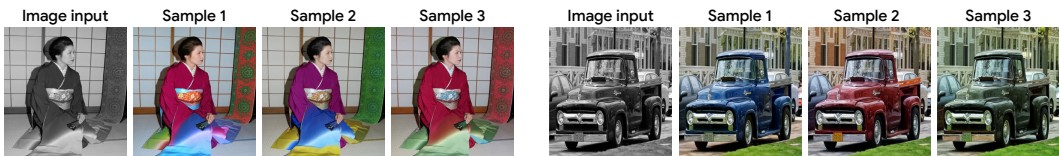

Figure 4: UViM outputs for the colorization task. Different samples produced by re-sampling the guiding code from the *language model* LM. Visually, the resulting samples are consistent and diverse.

## 3.2 Colorization

Colorization requires mapping grayscale pixels of an image to plausible colors. In particular for a given input there are many possible outputs and as so Fréchet Inception Distance (FID) [17] is a common metric. We opt to model this as a mapping from grayscale to RGB and use mean squared error as reconstruction loss during stage I training. For training we use ImageNet [39] training split consisting of 1.2M examples. We follow COLTRAN [28] and report FIDs using the prescribed splits of center-cropped images for metric computation and resize our model predictions to $256 \times 256$. We train stage I for 100 epochs and stage II for 50 epochs.

Figure 4 demonstrates that UViM is capable of producing high-quality and *diverse colorizations* for natural images. Table 1 shows that it achieves an FID of 16.986 on this task. This is slightly below the current state-of-the-art Palette [40] which uses diffusion models to cover a variety of tasks that output natural images. But significantly above COLTRAN [28] which uses a conditional autoregressive transformer to output a low resolution colorization followed by an upsampling model.

## 3.3 Monocular depth estimation

Depth prediction is a 3D scene understanding task, which requires mapping every pixel to a depth value (distance to the camera). We quantize the depth into buckets using 256 uniformly spaced bins, and use softmax cross entropy as reconstruction loss during stage I.

Table 2: Effect of ablating various UViM components on the panoptic segmentation task. PQ metric on holdout set. Besides stage II results (second row, black), we also show stage I using the restricted oracle's guiding code (first row gray). † ViT-L with 48 layers (+30% extra compute than UViM).

|  | Default | From Scratch | no Dropout | no Oracle | no Autoreg. | no Image |
|---|---|---|---|---|---|---|
| **UViM** (stage I) | 75.7 | 75.7 | **85.8** | 19.6 (25.5†) | 75.7 | 66.1 |
| **UViM** (stage II) | **43.7** | 41.4 | 42.2 | N/A | 33.3 | 39.1 |

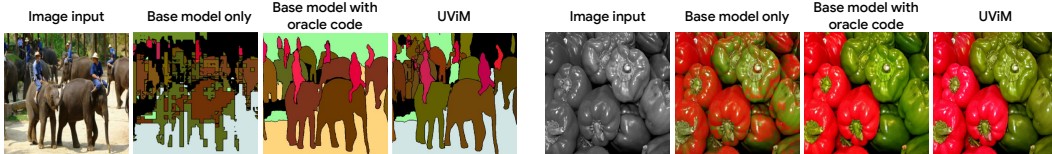

Figure 5: Outputs of various models in our ablation. We demonstrate that *base model* alone is not capable of modeling structured outputs, but when supported with compact oracle's guiding code, it achieves near perfect results. For completeness, we also present the result of the final UViM model.

We train on the NYU Depth V2 [43] dataset consisting of 47'584 training examples captured across 280 indoor scenes, and 654 official validation examples. For hyper-parameter selection we hold out all examples from 14 scenes from the training set. For evaluation, we report the common evaluation metric: root mean squared error (RMSE) on the standard crop of the evaluation images from [10]. At test time, we resize UViM outputs to the crop resolution via nearest neighbour. We train stage I for 200 epochs and stage II for 50 epochs.

Table 1 shows that UViM achieves an RMSE of 0.467 RMSE on this task. To contextualize this result, this score is comparable to DenseDepth [1] which uses an architecture composed of a pre-trained DenseNet-169 followed by upsampling layers and skip-connections. Our results still lags behind the most recent state-of-the-art model for this task [29] which consists of a mixed classification/regression loss, adaptive bins, auxiliary scene classification task and multi-scale prediction refining. However, UViM has very little task-specific tuning; our depth model is almost identical to out setup for panoptic segmentation, even sharing most hyperparameter values.

# 4 Ablations

In this section we dive deep into understanding UViM and perform ablations of its key components. We run extensive ablations (summarized in Table 2) on the panoptic segmentation task. For completeness, together with the performance of the final UViM models, we demonstrate the performance of UViM models after stage I training, which use the code from the restricted oracle. Some of our ablations are also illustrated visually in Figure 5.

For the default setting we follow the main experimental setup, but use $512 \times 512$ inputs for stage II training. To avoid overfitting to test data, all our ablations are performed using our custom splits, where we hold out 4096 randomly selected images from the training data and use those for evaluation.

**Ablating pre-trained weights.**

UViM is designed to make transfer learning easy, as it uses plain parametric models (without any modifications) that are commonly used for large-scale pre-training [46]. Nevertheless, we ablate the usage of pre-trained weights: "From Scratch" column in Table 2 shows the results (note that stage I results are not affected as we only use pre-trained weights for the LM). We use a longer training schedule with 500 epochs, which improves from-scratch results due to slower convergence. We observe

Table 3: PQ metric in holdout set to ablate effect of pre-trained weights and stage II model size.

|  | Base | Large |
|---|---|---|
| From Scratch | 41.0 | 41.4 |
| Pre-trained | 41.9 | 43.7 |

that the from-scratch trained model performs well and achieves competitive results only 2.3 PQ points behind the default setup of using pre-trained weights. Additionally, in Table 3, we ablate the scale of the pre-trained model. Notably switching the model size only provides 0.4 PQ points difference when training from scratch, but that difference is 1.8 PQ points when using pre-trained weights.

**Ablating code dropout.** The idea of the code dropout procedure, described in section 2.3, is to make the *guiding code* learned during stage I less "strict", so it will be easier to model it with the LM in stage II. Table 2 shows results of ablating this procedure in the "no dropout" column. As expected, ablating dropout results in better stage I results (by approximately 10 PQ points), as the oracle's code is not weakened by the dropout. On the other hand, the final UViM model becomes worse, as the resulting code learned without dropout has much more complex structure. We support our intuition by comparing final training losses of the LM models, trained for code with and without dropout. The losses are measured as average negative log-likelihoods, as are equal to $1.3$ and $4.2$, confirming that the code that was trained with dropout are much easier to learn. For the depth estimation task, we observed no difference ablating code dropout, indicating that code dropout is not always necessary, only for tasks where the code can become challenging for the LM to learn.

**Ablating restricted oracle model.** In this ablation we evaluate the base model $f : \mathcal{X} \mapsto \mathcal{Y}$ trained directly without $z$. The results in Table 2 confirm our (see Figure 5) qualitative assessment that directly predicting panoptic mask with pixel-wise loss works very poorly in the absence of the oracle model. Even when using a ViT-L with 48 layers as base model trained without $z$, which requires 30% extra compute than UViM, the performance is only $25.5$ PQ points.

**Ablating autoregressive structure of LM model.** So far we have assumed that the guiding code $z$ needs to predicted by a function capable to model a joint probability distribution, such as an autoregressive LM. We ablate this design choice and train a non-autoregressive LM, which predicts all components of $z$ in a single pass, but otherwise is identical to default LM models that we use.

The results in Table 2 confirm that the autoregressive component for joint probability distribution modeling is crucial. Ablating this component leads to a significant quality drop of $10.4$ PQ points. We observe a similar effect on depth estimation, where RMSE drops from 0.47 to 0.55.

**Ablating image input at stage I training.** One interesting ablation is to hide the image input from the base model $f$. In this case our whole UViM model can be interpreted from a different, more limited perspective: $\Omega$ learns to compress a label into $z$ and, at the same time, $f$ learns to decode it back into the original label. Then the LM learns to solve the task in this new learned compact space.

As shown in column "no image" of Table 2, the model solving the task in the learned space $\mathcal{Z}$ still performs reasonably well, though it lags behind the more general default approach. For depth estimation, the base model with no image obtains a similar performance to the full model (within $0.005$ RMSE), indicating that for this task the oracle can compress all of the information required to reconstruct the label into the guiding code.

**Varying oracle code length and dictionary size**

Finally, we investigate how the size of the code $z \in \mathcal{Z}$ affects performance. In particular, we vary the code length and dictionary size (the total number of discrete values for each component). Intuitively, a longer sequence and a larger dictionary make it easier to learn during stage I training, as the oracle "restriction" becomes weaker. However, it is not clear how the code parameters will affect the stage II training and the final UViM model, as longer sequence and more discrete values are potentially harder to learn for the LM model.

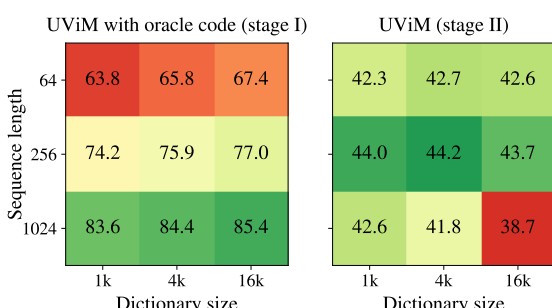

Figure 6: UViM model performance for the panoptic task (measured as PQ points).

To study this trade-off we train nine models: a cross-product of sequence lengths $\{64, 256, 1024\}$ and dictionary sizes $\{1024, 4096, 16384\}$. Figure 6 shows the results. As expected, UViM with oracle stage I model monotonically benefits from longer sequences and bigger dictionary sizes. However, the sweet spot for the final model is the code which is neither too long nor too short.

In Appendix C we additionally demonstrate outcome of the identical experiment for the image colorization task.

# 5  Related work

This paper is related to the vast amount of literature in computer vision, as the proposed modeling approach aims at unifying a wide array of vision tasks. We focus on the most related work that is either pushing in the same direction of model unification or uses highly related modeling techniques.

**Generative and autoregressive models.** Like in generative modeling, we have a similar goal of modeling high-dimensional structured outputs. A notable work, Pix2Pix [19], uses a conditional GAN model to map arbitrary image input to arbitrary image outputs. Despite going beyond generative tasks, and showing some outputs for semantic segmentation task, this model has not become a competitive approach, likely due to the complexity and instability of GAN training.

Autoregressive models gained a popularity in computer vision as (conditional) image generation tools [50, 49, 41] and later were used for tasks like image colorization [38, 14, 28]. However, scalability of autoregressive models for very high-dimensional outputs is a big problem, which was necessitating additional complexity, such as hierarchical generation [49, 25] or learning of an additional upsampling model [14, 28]. The idea of modeling a complex structured target by recurrent "autoregressive" invocations of a model was used in a customized implementations for visual relationship prediction [24] and human pose estimation [12].

Closer to our approach is the use of *learned discrete representations* with an autoregressively learned prior [51]. DALL-E [36] showed text conditioned image generation by using a decoder-only transformer to model a sequence of text and image discrete representations. VQGAN [11] show high-quality natural image generation conditioned in arbitrary image inputs by using an adversarial and perceptual loss to learn discrete representations. VIT-VQGAN [54] improved class-conditioned image synthesis with codebook improvements and parameterizing VQGAN with ViT [9]. Similarity NÜWA [53] propose a 3D transformer encoder-decoder, which covers language, image, and video with learned discrete representations. Notably, these works concentrate on the (conditional) generative image tasks and mostly ignore discriminative image tasks.

**Scene understanding.** There are several fundamental vision tasks that require a model to perform high-level scene parsing, such as object detection, instance or panoptic segmentation. Many standard methods, such as Faster-RCNN [37], Mask-RCNN [15] and RetinaNet [30] produce "dense" predictions for a large number of scored anchor boxes, followed by an ad-hoc non-maximal suppression procedure to eliminate redundant boxes. DETR [2] takes an alternative approach with an end-to-end model using a set-based global loss (via bipartite matching of proposals and ground truth). The DETR model can also be used for panoptic segmentation [23], where initial approaches involve combining models optimized for each sub-part of the task (instance and semantic classification). MaX-DeepLab proposes a box-free end-to-end approach that directly predicts class-labeled masks with a mask transformer. MaskFormer [6] further confirms that the mask classification view of the problem is important for semantic segmentation. Mask2Former [5] restricts the cross-attention learning around the predicted masks, leading to faster convergence and improved performance. Despite some promising convergence in the scene understanding area, the proposed approaches remain only viable for an important, but relatively narrow range of tasks.

**Vision model unification.** The *Perceiver IO* model [20] proposes an architecture that can efficiently process high-dimensional inputs and outputs, but, unlike UViM, it is not designed to model joint distribution of structured outputs. PIX2SEQ [4] proposes a model highly related to ours. It leverages a plain (sequence) language model for tackling the highly structured task of object detection. However, it is limited to the scenario when an output of a vision task can be manually represented as a short discrete sequence, which is rarely true for vision tasks. In [33] the authors propose a Transframer model, which uses a language model for modeling image outputs represented as sparse discrete cosine transform codes. However, the paper only shows qualitative results for "discriminative" tasks. Moreover, in comparison to our model, the Transframer is less flexible and powerful because it relies on the pre-defined fixed transform, while UViM learns discrete representations using a powerful end-to-end approach.

# 6  Conclusion and Discussion

UViM is a modeling approach for vision with an ambitious goal of unifying diverse vision tasks with one technique. Our resulting model consists of two components: an autoregressive language model (for modeling complex structured outputs) and a plain feed-forward base model that helps

to handle high dimensional outputs efficiently. Empirically, we confirm that UViM is capable of tackling diverse vision tasks in a unified way, while achieving competitive results. Our tasks cover semantic scene understanding (panoptic segmentation), conditional generative image modeling task (colorization) and 3D scene prediction (depth prediction).

Societal impact: General approaches, like UViM, could one day lead machine learning to be used more widely in settings where previously significant domain knowledge would be required and thus facilitate misuse or unintentional misspecification of a model. In particular when models are used to generate large outputs it is significantly harder to control those to stay within safe margins and to understand their impact when deployed.

We see UViM as a brave new prototype of the general-purpose learning approach for computer vision. As such, it still has many rough edges that need more research. We do not yet fully understand how to learn the optimal *guiding code*. Empirically, we observe that the final result is sensitive to the phase I code learning parameters. For example, code length of 256 seems overall better than 16 and 1024 in our experiments; or adding dropout to the code during its training results in a better final model. We hope future research will come up with better understanding of how to set up learning of the *guiding code*, beyond pure empirical observations. Another aspect is the computational and efficiency, which can be harder to control for the two-stage learning approach. More research may be needed to find design choices that will lead to much more efficient training procedures.

## Acknowledgments

We thank Ilya Tolstikhin, who was involved in the initial stages of the project and provided a useful feedback on UViM presentation. We thank Ting Chen for discussions at the early stage of this project and Liang-Chieh (Jay) Chen for the discussion on the panoptic segmentation task. Additionally, we thank Manoj Kumar who answered our questions on the evaluation for the colorization task. We also thank Daniel Keysers for feedback on the text of this paper. We thank anonymous reviewers for discussion in further experiments included in the final work.

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
