# A    Random sample of UViM predictions

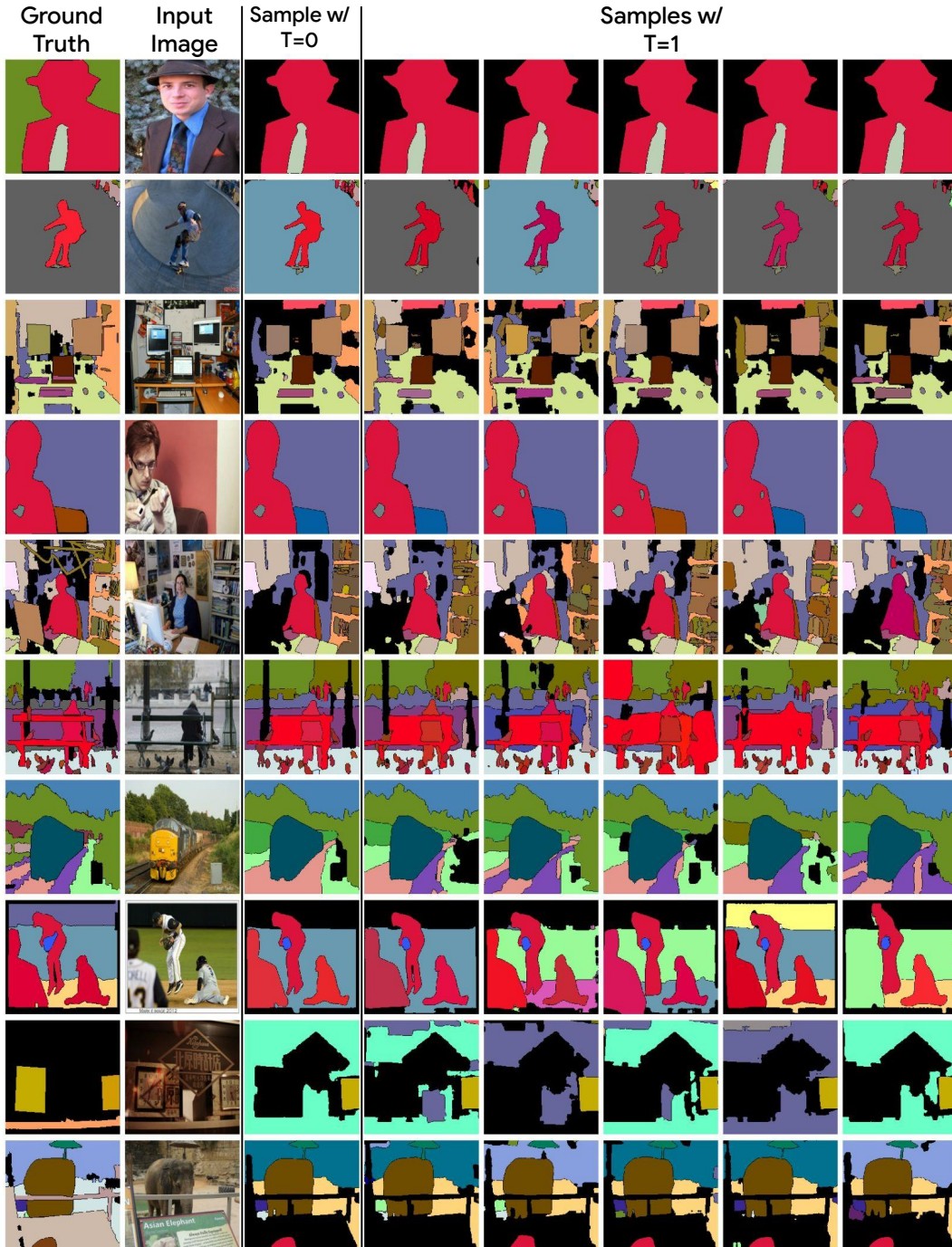

Figure 7: Random UViM example outputs for the panoptic segmentation task. The first column is the ground truth, second is the input image. The remaining columns are model outputs, sampling the *guiding code* with $T = 0$ (i.e. coordinate-wise argmax) in the third column, and with $T = 1$ in the remaining ones.

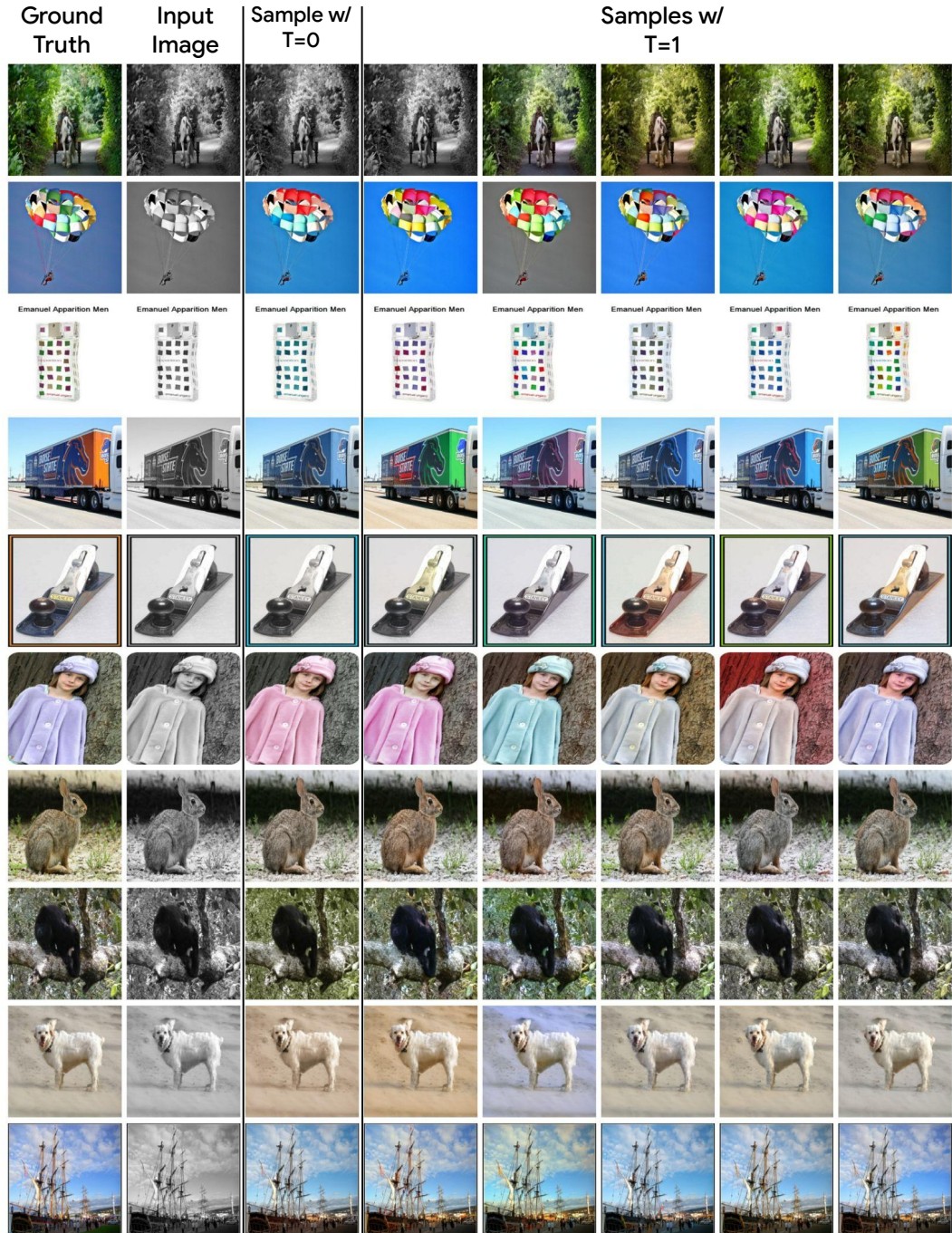

Figure 8: Random UViM example outputs for the colorization task. The first column is the ground truth, second is the grayscale input image. The remaining columns are model outputs, sampling the *guiding code* with $T = 0$ (i.e. coordinate-wise argmax) in the third column, and with $T = 1$ in the remaining ones.

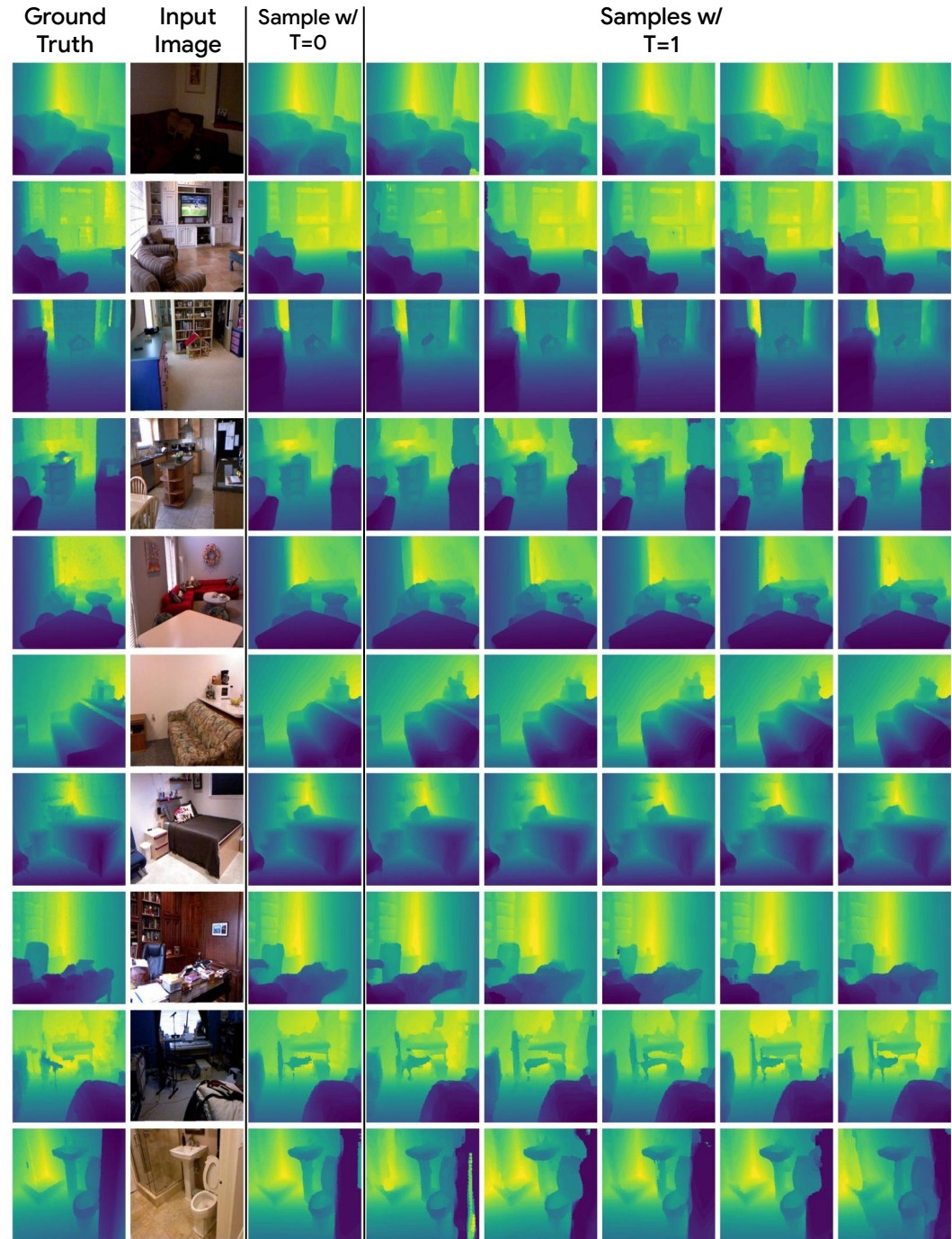

Figure 9: Random UViM example outputs for the depth prediction task. The first column is the ground truth, second is the input image. The remaining columns are model outputs, sampling the *guiding code* with $T = 0$ (i.e. coordinate-wise argmax) in the third column, and with $T = 1$ in the remaining ones.

## B    Masking guiding code

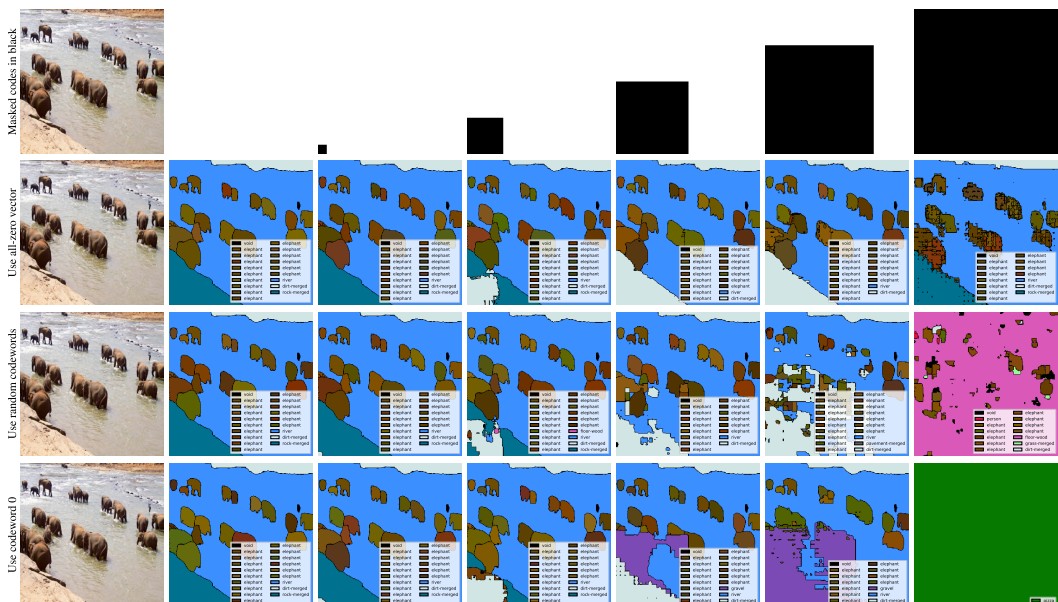

Figure 10: Best viewed on screen and zoomed in. To inspect whether the learned guiding code has a 2D-structure, we mask out a progressively larger region of the code predicted by the language model, as indicated by the black region in the first row. We then observe the effect this has on the base model's generated output. We show three different ways of masking codewords. **First**, we replace the embeddings of the codewords by (continuous) all-zero vectors, meaning the input to the base model consists of zeros. This result is consistent with Fig 1, showing the base-model can output reasonable, albeit inconsistent, results in the corrupted regions. **Second**, we replace the (discrete) codewords in the masked region by randomly sampled codewords. And **third**, we replace the (discrete) codewords in the masked region all by the same codeword. These last two rows show that, *when present, the base model puts high trust in the codewords coming from the language model*, even if they are inconsistent with the image pixels.

## C    Varying oracle code length and dictionary size for the colorization task

Figure 11 demonstrates FID colorization performance depending on the parameters of the guiding code. As expected, longer guiding code leads to better results for stage I models. Nevertheless, shorter codes lead to slightly better stage II performance.

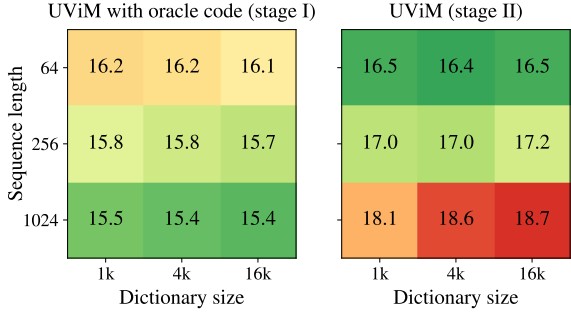

Figure 11: UViM model performance for the colorization task (measured as FID).

# D  Configuration files for the panoptic task.

This section demonstrates full configs with all hyper-parameters for training UViM stage I and II, which follow the `big_vision` codebase[3] conventions.

```python
1  RES = 512
2  PATCH_SIZE = 16
3
4  def get_config():
5    config = mlc.ConfigDict()
6
7    config.task = 'panoptic_segmentation'
8
9    config.dataset = 'coco/2017_panoptic'
10   config.val_split = 'train[:4096]'
11   config.train_split = 'train[4096:]'
12
13   config.batch_size = 1024
14   config.num_epochs = 1000
15
16   config.pp_train = (
17       f'decode|coco_panoptic|'
18       f'concat(["semantics","instances"], "labels")|'
19       f'randu("fliplr")|'
20       f'det_fliplr(key="image")|det_fliplr(key="labels")|'
21       f'inception_box|'
22       f'crop_box(key="image")|crop_box(key="labels")|'
23       f'resize({RES})|'
24       f'resize({RES}, key="labels", method="nearest")|'
25       f'value_range(-1, 1)|make_canonical|'
26       f'keep("image","labels")'
27   )
28
29   config.pp_eval = (
30       f'decode|coco_panoptic|'
31       f'concat(["semantics","instances"], "labels")|'
32       f'resize({RES})|'
33       f'resize({RES}, key="labels", method="nearest")|'
34       f'value_range(-1, 1)|make_canonical|'
35       f'keep("image","labels")'
36   )
37
38   config.shuffle_buffer_size = 25_000
39
40   config.log_training_steps = 50
41   config.log_eval_steps = 250
42   config.checkpoint_steps = 1000
43   config.keep_checkpoint_steps = 20_000
44
45   # Model section
46   config.model_name = 'proj.uvim.vit'
47   config.model = mlc.ConfigDict()
48   config.model.input_size = (RES, RES)
49   config.model.patch_size = (PATCH_SIZE, PATCH_SIZE)
50   config.model.code_len = 256
51   config.model.width = 768
52   config.model.oracle_depth = 6
53   config.model.base_model_depth = 12
54   config.model.mlp_dim = 3072
55   config.model.num_heads = 12
56   config.model.dict_size = 4096   # Number of words in dict.
57   config.model.codeword_dim = 768
58   config.model.dict_momentum = 0.995
```

---

[3]https://github.com/google-research/big_vision

```
59    config.model.with_encoder_ctx = True
60    config.model.with_decoder_ctx = True
61    config.model.inputs = {
62        # +1 for void label
63        'semantics': (133 + 1, PATCH_SIZE**2),
64        # COCO: actually 98 train/78 validation.
65        'instances': (100, PATCH_SIZE**2),
66    }
67    config.model.outputs = config.model.inputs
68
69    # Optimizer section
70    config.optax_name = 'scale_by_adafactor'
71    config.optax = dict(beta2_cap=0.95)
72
73    config.lr = 4e-4
74    config.wd = 4e-5
75    config.schedule = dict(decay_type='cosine', warmup_steps=4_000)
76    config.grad_clip_norm = 1.0
77
78    config.evals = [
79        ('panoptic_train', 'coco_panoptic'),
80        ('panoptic_holdout', 'coco_panoptic'),
81        ('panoptic_val', 'coco_panoptic'),
82    ]
83
84    base_eval = {
85        'pp': config.pp_eval.replace('decode|', ''),
86        'log_steps': 10_000,
87    }
88
89    config.panoptic_train = mlc.ConfigDict(base_eval)
90    config.panoptic_train.prefix = 'coco_panoptic_train/'
91    config.panoptic_train.split = 'train[4096:8192]'
92
93    config.panoptic_holdout = mlc.ConfigDict(base_eval)
94    config.panoptic_holdout.prefix = 'coco_panoptic_holdout/'
95    config.panoptic_holdout.split = 'train[:4096]'
96
97    config.panoptic_val = mlc.ConfigDict(base_eval)
98    config.panoptic_val.prefix = 'coco_panoptic/'
99    config.panoptic_val.split = 'validation'
100
101   return config
```

Listing 1: Full config for panoptic stage I training.

```python
LM_MODELS = {
    'base': dict(num_layers=12, num_heads=12,
                 mlp_dim=3072, emb_dim=768),
    'large': dict(num_layers=24, num_heads=16,
                  mlp_dim=4096, emb_dim=1024),
}
STAGE_I_MODELS = {
    'base': dict(oracle_depth=6, base_model_depth=12,
                 num_heads=12, mlp_dim=3072, width=768),
}
RES = LABEL_RES = 512
PATCH_SIZE = LABEL_PATCH_SIZE = 16

def get_config():
  config = mlc.ConfigDict()
  config.pp_modules = ['ops_general', 'ops_image', 'proj.uvim.pp_ops']

  config.pp_train = (
      f'decode|coco_panoptic|'
      f'concat(["semantics","instances"], "labels")|'
      f'randu("fliplr")|'
      f'det_fliplr(key="image")|det_fliplr(key="labels")|'
      f'inception_box|'
      f'crop_box(key="image")|crop_box(key="labels")|'
      f'resize({LABEL_RES}, inkey="image", outkey="image_ctx")|'
      f'resize({RES})|'
      f'resize({LABEL_RES}, key="labels", method="nearest")|'
      f'value_range(-1, 1, key="image_ctx")|'
      f'value_range(-1, 1)|make_canonical|'
      'keep("image", "image_ctx", "labels")'
  )
  config.pp_eval = (
      f'decode|coco_panoptic|'
      f'concat(["semantics","instances"], "labels")|'
      f'resize({LABEL_RES}, inkey="image", outkey="image_ctx")|'
      f'resize({RES})|'
      f'resize({LABEL_RES}, key="labels", method="nearest")|'
      f'value_range(-1, 1, key="image_ctx")|'
      f'value_range(-1, 1)|make_canonical'
      f'|keep("image", "image_ctx", "labels")'
  )
  pp_predict = (
      f'resize({LABEL_RES}, inkey="image", outkey="image_ctx")|'
      f'resize({RES})|'
      f'value_range(-1, 1, key="image_ctx")|'
      f'value_range(-1, 1)|'
      f'keep("image", "image_ctx", "image/id")'
  )

  config.dataset = 'coco/2017_panoptic'
  config.train_split = 'train[4096:]'
  config.val_split = 'train[:4096]'
  config.shuffle_buffer_size = 50_000
  config.batch_size = 512
  config.num_epochs = 200

  config.log_training_steps = 50
  config.log_eval_steps = 1000
  config.checkpoint_steps = 1000
  config.keep_checkpoint_steps = 5000

  # Optimizer section
  config.optax_name = 'scale_by_adafactor'
  config.optax = dict(beta2_cap=0.95)
  config.lr = 0.001
```

```python
66    config.wd = 0.000001
67    config.lr_mults = [
68        ('pos_embedding_encoder.*', 0.1),
69        ('EmbedPatches.*', 0.1),
70        ('encoder.*', 0.1),
71        ('decoder.*', 1.0)
72    ]
73    config.schedule = dict(decay_type='cosine', warmup_steps=4_000)
74
75    # Restricted oracle section
76    config.oracle = dict()
77    config.oracle.task = 'panoptic_segmentation'
78    config.oracle.model_init = 'oracle.npz'
79    config.oracle.model_name = 'proj.uvim.vit'
80    config.oracle.model = STAGE_I_MODELS['base']
81    config.oracle.model.input_size = (LABEL_RES,LABEL_RES)
82    config.oracle.model.patch_size = (LABEL_PATCH_SIZE,LABEL_PATCH_SIZE)
83    config.oracle.model.code_len = 256
84    config.oracle.model.dict_size = 4096
85    config.oracle.model.codeword_dim = 768
86    config.oracle.model.with_encoder_ctx = True
87    config.oracle.model.with_decoder_ctx = True
88    config.oracle.model.inputs = {
89        # +1 for void label
90        'semantics': (133 + 1, LABEL_PATCH_SIZE**2),
91        # COCO: actually 98 train/78 validation.
92        'instances': (100, LABEL_PATCH_SIZE**2)}
93    config.oracle.model.outputs = config.oracle.model.inputs
94
95    # Model section
96    config.model_name = 'proj.uvim.lm'
97    config.model_init = {'encoder': 'howto-i21k-L/16'}
98    config.model = LM_MODELS['large']
99    config.model.patches = dict(size=(PATCH_SIZE, PATCH_SIZE))
100   config.model.vocab_size = config.oracle.model.get_ref('dict_size')+1
101   config.model.posemb_type = 'learn'
102   config.model.input_size = (RES, RES)
103   config.model.seq_len = config.oracle.model.get_ref('code_len')
104
105   # Evaluation section
106   config.evals = [
107       ('panoptic_train', 'coco_panoptic'),
108       ('panoptic_holdout', 'coco_panoptic'),
109       ('panoptic_val', 'coco_panoptic'),
110   ]
111   base_eval = dict(pp=pp_predict, log_steps=10_000)
112
113   config.panoptic_train = dict(base_eval)
114   config.panoptic_train.prefix = 'coco_panoptic_train/'
115   config.panoptic_train.split = 'train[4096:8192]'
116
117   config.panoptic_holdout = dict(base_eval)
118   config.panoptic_holdout.prefix = 'coco_panoptic_holdout/'
119   config.panoptic_holdout.split = 'train[:4096]'
120
121   config.panoptic_val = dict(base_eval)
122   config.panoptic_val.prefix = 'coco_panoptic/'
123   config.panoptic_val.split = 'validation'
124
125   return config
```

Listing 2: Full config for panoptic stage II training.