# OpenReview forum: "UViM: A Unified Modeling Approach for Vision with Learned Guiding Codes"
_NeurIPS.cc/2022/Conference — NeurIPS 2022 Accept_

### Official Review · Reviewer_bsWT · 2022-06-20

**Rating:** 8
**Confidence:** 3
**Soundness:** 4 excellent
**Presentation:** 3 good
**Contribution:** 3 good

**Summary:**

This paper presents an approach for computer vision tasks involving large output spaces overlaying the image -- like colorization, object detection, etc. A challenge with these tasks is that, because they have large output spaces, it is nontrivial to model them in an autoregressive setting, even though such a setting works great for NLP tasks. This paper proposes to learn a "guiding code" that represents the image overlay over a latent space of discrete tokens. The authors use 256 tokens in total, so the discrete tokens are highly compressed versus the [H x W x C] overlay.

The paper proposes a two-stage approach for training. In the first stage, the model learns the code through a VQ-VAE like setting. A 'restricted oracle' is used that gets access to the labels and must summarize it through the discrete code; the 'base model' must reconstruct the labels given the code and the input image. During the second stage, the model learns to generate the guiding codes given images (mimicing the oracle given just the image).

The paper shows strong results across a variety of different tasks, including Panoptic Segmentation (COCO), NYU Depth, and ImageNet colorization. The paper has helpful ablations that discusses what helps and doesn't in the proposed framework, including what the optimal sequence length is.


**Questions:**


* Is there some way to visualize the codes to get a sense of what the model is doing?
* Could it be possible for the approach (given that it is to some degree class-agnostic) to generalize even beyond the supervised data sources that it's finetuned on?

**Limitations:**

The social impact section feels very weak to this reviewer. The following is not a strong argument: "the proposed modeling approach is very general ... As a result, we do not anticipate any negative societal impact of our technique."

There is a limitations section, but it might be great to speculate a bit more about what current bells and whistles are truly needed in other NMS/Hungarian Algorithm detection/segmentation approaches, and where this approach might fit in.

I'd be willing to adjust my score if these points are better handled.

**Strengths And Weaknesses:**


Strengths:
* This paper presents a very simple approach that brings together several different 'image overlay' tasks from computer vision. To the best of this reviewer's knowldege, this model is novel and creative. It will likely be of interest to many in the community given the recent successes of image generation approaches like DALL-E.
* The results are strong, especially for not being very optimized for these diverse vision tasks.
* The ablations are helpful for understanding what the model is doing, showing that (among other things) the autoregressive model helps a lot, and there is a 'sweet spot' in terms of code length.

Weaknesses: An exceptionally weak limitations / social impact section (see later), but nothing else that stands out to this reviewer

---

> ### Author Response · Authors · 2022-08-02
> **Reply to Reviewer bsWT**
>
> > Visualize the codes
>
> Good question. While answering reviewer 8vD5, we produced a visualization to investigate if the codes have a 2d-structure, e.g. local modifications have local impact. Additionally checkpoints and notebooks will be made available.
>
> > Generalize beyond supervised data
>
> We are interested in exploring such directions, but have nothing to share at the moment.
>
> > Societal impact
>
> Given the early stage it is hard to predict negative social impact from this specific work, however we will add the following text to highlight potential issues with general models and generation of large outputs:
>
> General modeling approaches, like UViM, could one day lead machine learning to be used more widely in settings where previously significant domain knowledge would be required and thus facilitate misuse or unintentional misspecification of a model. In particular when models are used to generate large outputs it is significantly harder to control those to stay within safe margins and to understand their impact when deployed.
>
> > Limitations
>
> We already discuss why the additional “bells and whistles” are needed in section 2, here we paraphrase the discussion: complexity of many vision tasks (e.g. panoptic segmentation) stems from the necessity to model structure of multi-dimensional outputs. As vision tasks often have very high-dimensional outputs, many general-purpose models that are capable to model structured outputs are computationally prohibitive. Thus, the currently dominating SOTA models typically resort to hand-crafted components on top of neural networks that produce outputs that respect the task structure. For example, for tasks like panoptic segmentation the current SOTA models operate on the level of objects and thus require extra components, such as matching losses or NMS. In contrast, UViM leverages a two-stage procedure which enables a general-purpose model for structured outputs to be effectively scaled to very high-dimensional outputs.
>
> As for more limitations, we will additionally mention potential challenges with data efficiency and model flexibility when it comes to “fully convolutional” application, as discussed above with the reviewer 8vD5.

---

> > ### Comment · Reviewer_bsWT · 2022-08-08
> > **thanks for the response!**
> >
> > Thanks -- this addressed my main concerns, so I updated my score from 7 to 8.
> >
> > After reading all the reviews (even 1EyD's more critical review) I think this is a strong paper that should be accepted. I feel persuaded by the authors' response to that review. The experiment comparing this method to the ViT-B baseline could easily be added to this paper, which should address that concern and make the paper stronger.

---

> > > ### Author Response · Authors · 2022-08-09
> > > **Thank you**
> > >
> > > Indeed, we agree the new experiments arising from the reviewer discussions are valuable additions, which we will include in the final paper version.

---

### Official Review · Reviewer_1EyD · 2022-07-08

**Rating:** 7
**Confidence:** 4
**Soundness:** 3 good
**Presentation:** 3 good
**Contribution:** 3 good

**Summary:**

The paper proposes UViM, a unified modeling approach for vision tasks with learned guiding codes. The approach has two stages. It first learns a discrete code from the GT to guide the prediction, and try to predict the code from the input in the second stage. Experiments on three dense vision tasks are conducted to prove the effectiveness and generalization of the method.

**Questions:**

I have no more questions.

**Limitations:**

The limitations have been well addressed in the paper.

**Strengths And Weaknesses:**

Strengths:

The idea is quite novel. Previously, the discrete code is mainly used for generative tasks, but in this work, it serves as a bridge between input and GT. The code is informative enough to guide the inference, since it has access to the GT in the first stage. The discrete bottleneck makes sure that the code is compact and abstract, so that it can be learned by a language model in the second stage. The autoregressive language model is flexible, making it a good choice to model the different structure of each task.

The paper is also neatly written. It is easy to follow.

Weakness:

Even though the idea is cool, and it may work well, I don't think the experiment is enough to support the effectiveness of the method. A unified structure for vision tasks has been studied for a while. Methods like Perceiver-IO and the following works have been trying to unify vision tasks by exploiting the flexibility of transformer structure. I think a comparison with Perceiver-IO or similar works should be presented.

The ablation study on restricted oracle model can compensate for the missing comparison, since the base model (f: X->Y) is already a vanilla baseline of unified structure. But the ablation is not fair. The LM (ViT-L/16 + BERT-Base) is much larger than the base model (ViT-B/16). It's hard to say whether the proposed method is better or it simply has more trainable parameters.

I feel sorry for having to give the paper a lower rating than the method should deserve, but the experiments can not convince me that the method is better than a vanilla baseline. If the missing experiments or a fair comparison with the base model is provided, I feel happy to change my rating.

---

> ### Author Response · Authors · 2022-08-02
> **Reply to Reviewer 1EyD**
>
> > Larger base model as the baseline
>
> Thank you for pointing this out. We perform an additional experiment and train the largest model that still fits the memory: ViT-L/16 with 48 layers (see comments below about compute). The results are better than the ViT-B baseline: the bigger model archives 25.5 PQ points, but it still falls strongly behind the comparable UViM model, which achieves 39.8 PQ points (from scratch baseline).
>
> The baseline ViT-L(layers=48, tokens=1024) model requires ~30% more flops than the UViM model, which is ViT-L(layers=24, tokens=1024) + ViT-L(layers=24, tokens=256) + ViT-B(layers=12, tokens=1024).
>
> Please also note that on a high level, the base model (or PerceiverIO model) lacks the ability to model the joint distribution of structured outputs and this can not be solved with scale. We illustrate the consequence of this in Figure 5. Even if per-pixel predictions are plausible (e.g. red/green bell peppers), the base outputs all pixel colors independently and results in the flickering effect.

---

> > ### Comment · Reviewer_1EyD · 2022-08-08
> > **Re: Rebuttal**
> >
> > Thank you for your response. My main concern is addressed. I would like to change the score to 7.

---

> > > ### Author Response · Authors · 2022-08-09
> > > **Thank you**
> > >
> > > Thank you very much for the positive reaction to our additional experiment. We will include it in the final paper version, as the capacity comparison is a valid concern.
> > >
> > > Please don't forget to update (edit) your original review with the new score before the end of the discussion period, such that it is properly indicated in the system.

---

### Official Review · Reviewer_8vD5 · 2022-07-11

**Rating:** 7
**Confidence:** 4
**Soundness:** 3 good
**Presentation:** 4 excellent
**Contribution:** 3 good

**Summary:**

This paper presents UViM, a "unified modeling approach for vision using learned guiding codes". More specifically, the main motivation of the paper is to unify the highly specialized and task-specific architectures that are present in typical computer vision tasks such as segmentation, depth estimation etc. To this end, the paper proposes a two-stage procedure inspired from NLP, where in the first stage a simple feedforward model $f: (x, y) \mapsto y$ with a discrete bottleneck in the $y$ data stream is learned. Here, $y$ denotes the target modality, e.g. a segmentation map, and $x$ is a natural image. In the second stage, the distribution of guiding codes is learned using a standard language model (encoder-decoder transformer), such that the final prediction model $f: (x, LM(x)) \mapsto y$ does only depend on an input image $x$. The model is trained on three different tasks (colorization, monocular depth estimation and panoptic segmentation) and achieves reasonable performance on all of these.

**Questions:**

Please see the above section on "strengths and weaknesses", mainly the section with mixed comments.

**Limitations:**

Yes, limitations are briefly discussed in the conclusion ("rough edges"), and no negative societal impact is anticipated.

**Strengths And Weaknesses:**

**Strengths:**

- This is a good paper: the motivation to unify models for different visual tasks is reasonable, and the approach presented here is a good step in this direction. It is very well written and easy to follow thanks to its good structure. I agree with the authors' statement that UViM is a "brave new prototype of the general-purpose learning approach for computer vision." (l.355).
- The paper performs lots of informative ablation studies (Sec.4), although some open questions remain.

**Weaknesses:**

- A drawback of the method is that it requires paired training data, and the LM backbone has to be trained on discrete representations of the target modality (which requires pixelwise supervision). Besides the colorization task, these supervisory data is usually expensive to gather, which limits the size of the training set. Since the latent representation is modeled with an autoregressive language model, this approach is prone to overfitting (is this why dropout on the representations is required?). It would be interesting to try equally general, but potentially more data-efficient approaches (e.g., discrete diffusion models [1,2]) here.
- l.127-128, "For historical reasons we equip Ω with an additional input x, though it appears not to affect the resulting model" are unclear to me. Does this mean that the final model is always trained with this stream but it does not hurt to remove it?
- Is the dependency on guiding code capacity (controlled via sequence length and codebook size) the same across modalities (cf Fig.6 which demonstrates this for the panoptic segmentation task only)
- Although performance is good, the approach does not yet reach that of more specialized model designs. I am not sure whether performance will reach that of more specialized models without using more specialized strategies to decode into the output space or architectural choices (which would argue against the ground tone of this work of "unifying" different areas)

**Mixed Comments:**
- For the colorization task: To increase fidelity, adding a (patch-wise) GAN loss to stage I training should help to improve FID and perceptual quality, similar to VQGAN.
- Does it help to retain spatial structure in the guiding codes? Since the model deals with image data, using 2D backbones to model the distribution of guiding codes could make the approach more applicable to smaller datasets.
- Towards a "true" unification: Would it be possible to train the LM on all domains simultaneously, potentially making use of cross-domain interdependencies, and use a "task-token" to signal which task the LM has to sample from?
- Does, similar to classical vision models,  a "convolutional application" work out-of-the-box? E.g., can we apply the model on larger resolution than it was originally trained on? I think this depends highly on the implementation of the positional encodings of the ViT backbone, would be good to add some explanation here.
- How important is the scale of the pre-trained encoder (the only comparison is to "from-scratch") ? Can it be fixed?
- Side question: Am I right that the model can be interpreted as a variant of "ViT-VQGAN" (with task-specific reconstruction loss) which is trained on "labels" instead of images but conditioned on the latter?

In summary, this is a solid paper, with  room for exploration and improvements in the future. I am recommending it for publication at the conference.


_References_
- [1]: Argmax Flows and Multinomial Diffusion: Learning Categorical Distributions, Hoogeboom et al
- [2]: Structured Denoising Diffusion Models in Discrete State-Spaces, Austin et al

---

> ### Author Response · Authors · 2022-08-02
> **Reply to Reviewer 8vD5**
>
> > Limited amount of pixel-wise training data and overfitting.
>
> We agree it is a valid concern. However, other specialized models are also not immune to this issue and the majority of previously developed techniques to mitigate overfitting are also applicable to UViM. This includes specialized augmentations [1], regularizations [2], pseudolabeling [3,4] and transfer from large datasets with rich labels such as OpenImages and Objects365. We also agree that data-efficient modeling of “guiding codes” is an interesting open question specific to UViM and hope future research will address this topic.
>
> > Additional image input to the oracle.
>
> Yes, your understanding is correct. We perform an additional ablation with our final setup for panoptic task to evaluate the importance of providing an image input to the oracle. The resulting models, with and without image input to oracle, achieve 44.2 PQ points on the holdout validation set. This confirms that image input is irrelevant for the final performance.
>
> > “How important is the scale of the pre-trained encoder?”
>
> We additionally ran stage II panoptic model with the pretrained ViT models of 4 sizes: Tiny, Small, Base and Large, using the publicly released models from [5]. The results are 8.8 PQ, 23.3 PQ, 41.9 and 44.2 PQ points respectively. These results confirm the benefit of using large pretrained models.
>
> > Code capacity importance for tasks beyond panoptic.
>
> We additionally explored how code capacity influences colorization tasks. The colorization FID results [ https://imgur.com/a/rcYsBKW ] show that this task benefits from a shorter code of length 64. This suggests that different tasks will generally benefit from different code settings, as panoptic worked the best with code length of 256. On the other hand, we observe that reasonably deviating from the per-task optimal setting does not result in a too steep performance drop, so sticking to a certain default value is a viable option and, if extra compute is available, one can further select better code parameters.
>
> > GAN loss for colorization task.
>
> We agree, this is one of the natural ways to potentially improve UViM for certain tasks, which is a viable direction for future research.
>
> > Structure of the guiding code
>
> Indeed, retaining 2D structure can be a desirable inductive bias for training on the small scale data, and can also enable “convolutional application” (see the discussion below). To this end we demonstrate that our current model already retains a certain amount of 2d structure in the code. In image [ https://imgur.com/a/mnLSqv7 ], we demonstrate that when we progressively mask the codes in a bottom left corner of the image, the distortions in the output are spatially aligned with the masking.
>
> We believe this happens due to the presence of skip-connections in the transformer model. CNNs are likely to retain such structure even stronger. Nevertheless, in our preliminary experiments we did not get good results with CNNs and therefore committed to use ViTs.
>
> > “Convolutional application”
>
> ViT backbones can be adapted to process input images of different resolution (with and without additional finetuning) via interpolation of positional embedding [6]. In fact, our flagship panoptic model uses image encoder pretrained on 224x224 resolution, while we apply it to 1280x1280 input images. Adjusting output resolution dynamically is more challenging with UViM, as the output resolution is tied to the code length. On the other hand, if the code retains 2D structure reasonably well, one can upsample in order to increase output resolution, though to make it work well additional investigation is likely needed.
>
> > “True unification”
>
> We do not see any conceptual roadblocks towards implementing a stronger notion of unification with UViM. Task-conditional unification as you describe is one option, while learning a joint code to represent multiple tasks is another one. We plan to further explore it in the future.
>
> > Relation to VQGAN
>
> This is indeed a highly related approach, which is similar to UViM in spirit. The notable difference is that we do not employ additional GAN loss, devise a custom VQVAE dictionary learning procedure that works well for classical vision applications and, as you mention, we devise a mechanism for conditioning on the extra (image) input (for both stage I and stage II models).
>
>
> [1] Simple Copy-Paste is a Strong Data Augmentation Method for Instance Segmentation by Ghiasi et al.
>
> [2] Deep Networks with Stochastic Depth by Huang et al.
>
> [3] Rethinking Pre-training and Self-training by Zoph et al.
>
> [4] Multi-Task Self-Training for Learning General Representations by Ghiasi et al.
>
> [5] How to train your ViT? Data, Augmentation, and Regularization in Vision Transformers by Steiner et al.
>
> [6] An Image is Worth 16x16 Words: Transformers for Image Recognition at Scale by Dosovitskiy et al.

---

> > ### Comment · Reviewer_8vD5 · 2022-08-07
> > **Re: Rebuttal**
> >
> > Hi, thank you for your detailed responses and additional analysis. I think including these in the paper would make it even stronger, especially the section on the spatial structure of the guiding code. I stand by my original rating of 7 and recommend the paper for publication.

---

> > > ### Author Response · Authors · 2022-08-09
> > > **Thank you**
> > >
> > > Thank you for your questions, we believe they are valuable and we will indeed add the new experiments and discussion points to the final paper.

---

### Meta-Review · Area_Chair_yw7B · 2022-08-25

**Recommendation:** Accept
**Confidence:** Certain

**Metareview:**

Three reviewers provided positive reviews which were further strengthened post discussion. They agreed that that the motivation was strong, the model was novel and the paper was well written. They appreciated the ablations provided by the authors and found the results compelling. The main concern by the reviewers was a missing experiment which was provided by the authors in their rebuttal, and was appreciated by multiple reviewers. In summary, the reviewers are unanimous in their support of this paper. I agree with their reviews and I recommend acceptance.

**Award:**

No

---

### Decision · Program_Chairs · 2022-09-14

Accept